# A Systematic Survey on the Role of Cloud, Fog, and Edge Computing Combination in Smart Agriculture

**DOI:** 10.3390/s21175922

**Published:** 2021-09-03

**Authors:** Yogeswaranathan Kalyani, Rem Collier

**Affiliations:** School of Computer Science, University College Dublin, Belfield, D04 V1W8 Dublin, Ireland; kalyani.yogeswaranathan@ucdconnect.ie

**Keywords:** cloud, fog, edge, smart agriculture

## Abstract

Cloud Computing is a well-established paradigm for building service-centric systems. However, ultra-low latency, high bandwidth, security, and real-time analytics are limitations in Cloud Computing when analysing and providing results for a large amount of data. Fog and Edge Computing offer solutions to the limitations of Cloud Computing. The number of agricultural domain applications that use the combination of Cloud, Fog, and Edge is increasing in the last few decades. This article aims to provide a systematic literature review of current works that have been done in Cloud, Fog, and Edge Computing applications in the smart agriculture domain between 2015 and up-to-date. The key objective of this review is to identify all relevant research on new computing paradigms with smart agriculture and propose a new architecture model with the combinations of Cloud–Fog–Edge. Furthermore, it also analyses and examines the agricultural application domains, research approaches, and the application of used combinations. Moreover, this survey discusses the components used in the architecture models and briefly explores the communication protocols used to interact from one layer to another. Finally, the challenges of smart agriculture and future research directions are briefly pointed out in this article.

## 1. Introduction

Agriculture plays a fundamental role in the world both as a key source of livelihood and its role in the global food supply chain. It is a foundation of human survival. However, factors such as population growth, expansion of industrial development, and climate change all restrict agricultural development [1]. The UN has projected that the world’s population will reach 9.7 billion by 2050 [2]. If these projections materialise, annual world agricultural production would need to increase by some 60 percent between 2010 and 2050 [3]. To meet this demand, farmers, scientists, agronomists, and agricultural industries turn to new technologies, such as Cloud Computing, Edge Computing, Fog Computing, IoT, Big Data, Artificial Intelligence, and Drones.

In recent years, concepts such as smart farming, smart agriculture, or precision agriculture have become more popular [4,5,6]. These concepts are generally regarded as the same, and the terms can be used interchangeably. Smart farming uses the new technologies in the agricultural domain to make maximum use of resources and minimise the environmental impact. Currently, sensors can offer highly accurate measurements of crop status. Based on those values, actuators can manage agricultural processes related to animals, crops, greenhouses, irrigation, soil, and weather. This can result in improvements to harvest forecasting, weather prediction, increase production, water conservation, real-time data collection, and production, lowered operation costs, equipment monitoring, remote monitoring, and accurate farm and field evaluation.

Smart agriculture is always connected with high volumes of heterogeneous data sources such as autonomous tractors, harvesters, robots and drones, sensors, and actuators. Heterogeneous sensors and other devices collect relevant agricultural data such as humidity, temperature, pH, and soil conditions. Similarly, it considers the use of various actuators, such as water sprinklers, ventilation devices, lighting, automated windows (in glasshouses), and soil and water nutrition pumps that react according to the data. The number of cloud-based agricultural standalone systems and physical systems is increasing on an almost daily basis, helping to achieve a range of monitoring and analysing objectives.

Moreover, the last few years of publications have shown that modern computing paradigms such as Cloud, Fog, and Edge play a vital role in agriculture. The main applications of Cloud, Fog, and Edge Computing in agriculture are crop farming, livestock, and greenhouses, which are grouped into different application domains. Some of these applications are implemented with the help of IoT-based sensors and devices by using wireless sensor networks (WSNs), and some other applications are developed with combinations of new computing. For instance, Cloud and Fog, Cloud and Edge, Fog and Edge, or Cloud–Fog–Edge and IoT. Therefore, it is essential to collect, summarise, analyse, and classify state-of-the-art research.

This review aims to identify how Cloud–Fog–Edge combinations are used with smart agriculture by addressing the following questions such as (1) How are new technologies such as Cloud, Fog, and Edge used in smart agriculture, and what features of agriculture are covered? (2) What components are used in the architecture? (3) What type of combinations of computing is used? (4) What are the future direction and opportunities for smart agriculture using Cloud–Fog–Edge Computing? The key contributions of this review relating to smart agriculture and Cloud, Fog, and Edge computing are outlined as follows. Section 2 presents a summary of basic concepts such as Cloud Computing, Fog Computing, Edge Computing, Smart Agriculture, and Edge vs. Fog, which are required background knowledge to this review. Section 3 discusses the latest studies on smart agriculture and our motivation for this research to give a more up-to-date view to the readers about smart agriculture and Cloud–Fog–Edge. Section 4 presents research methodology by defining research questions and objectives, search queries, and databases. The core contribution of this research is contained in Section 5, where it summarises the details of findings with regard to application domains, research approaches, existing applications, and combinations used in these applications. Section 6 briefly explores the challenges of smart agriculture and future direction. Finally, Section 7 provides conclusions of the article.

## 2. Background

This section summarises fundamental concepts such as smart agriculture, Cloud Computing, Fog Computing, and Edge Computing. Additionally, it gives a high-level idea to the readers to understand the background and questions related to Cloud, Fog, and Edge Computing in smart agriculture.

### 2.1. Smart Agriculture

Smart Agriculture or Smart Farming is an emerging concept that uses modern technologies in agriculture and livestock production to increase production, quantity, and quality, by making maximum use of resources and minimising the environmental impact. This is demonstrated when farmers and all stakeholders related to agriculture use modern technologies and smart devices to monitor their farms, equipment, crops, and livestock. Using these devices, they can also obtain statistics on their livestock feeding and production of crops [5,7,8].

In recent years, smart farming has become helpful to all agricultural stakeholders from small to large scale. Smart farming provides benefits not only to scientists and agronomists but also farmers to access modern technologies and devices that help in the maximization of product quality and quantity while reducing the cost of farming [5]. Smart farming mainly focuses on soil fertility, energy, grassland, water, feed, inputs and waste, machinery, and time management [9,10].

The integration of modern technologies with agriculture achieves the objectives of smart agriculture such as efficiency, sustainability, and availability [11], increased production, water-saving, better quality, reduced costs, pest detection, and animal health [12,13]. The other aims are to increase the reliability of spatially explicit data [5], make agriculture more profitable for the farmer [5], and offer the farmer the option of actively intervening in processes or controlling them [14]. Moreover, big data analysis is another goal of smart farming. Big data consist of massive volumes and a wide variety of data that are generated and captured by agricultural sensors and actuators. In particular, data collected from the field, farm, animals, and greenhouses include information on planting, spraying, materials, yields, in-season imagery, soil types, and weather. Big data analysis provides efficient techniques to do a quality analysis for decision-making [4]. In the coming years, smart agriculture is projected to create a significant impact on the world agricultural economy by applying all modern technologies.

### 2.2. Cloud Computing

Nowadays, Cloud Computing represents the most advanced computing paradigm. According to [15], the term “Cloud Computing” was first used by Google and Amazon in 2006. In 2019, Ref. [16] offered the latest definition of Cloud describing it as *a computing paradigm for providing anything as a service such that the services are virtualised, pooled, shared, and can be provisioned and released rapidly with minimal management effort*. The Cloud is composed of five characteristics, three service models, and four different deployment models [17]. Cloud Computing offers key services such as infrastructure as a service (IAAS), platform as a service (PAAS), and software as a service (SAAS) [18]. The four deployment models are private Cloud, Community Cloud, Public Cloud, and hybrid Cloud [17]. The characteristics of Cloud are on-demand self-service, broad network access, resource pooling, rapid elasticity, and measured service [17]. Even though Cloud Computing has benefits (cost savings, efficiency, scalability, reliability), it also has some challenges when it deals with a massive amount of data. For example, low latency, high internet bandwidth, real-time analytics, data management, load balancing, energy consumption, security, and privacy are some challenges of Cloud Computing. Moreover, most of the computations happen directly in the Cloud as it is a centralised computing model. In the last few decades, Cloud Computing played a significant role in the agriculture field. Data acquisition and remote storage, low-cost access to ICT resources, online agriculture experts consultation, land records automation, and weather forecasting are the main features of Cloud Computing in agriculture [19]. Similarly, constant and high-speed network connectivity, security, and privacy are the challenges in the use of Cloud in the agricultural domain [19].

### 2.3. Fog Computing

The term ‘Fog Computing’ was first mentioned and defined by Flavio Bonomi at CISCO in 2012. According to [20], *Fog Computing is a highly virtualised platform that provides compute, storage, and networking services between end devices and traditional Cloud Computing data centers, typically, but not exclusively located at the edge of network*. Thus, the user’s computation demand is served at their proximity rather than performing it in the distance Cloud. Moreover, Fog Computing is primarily introduced for applications that need real-time processing with low latency. The Fog layer is composed of large-scale geo-distributed Fog nodes, which are deployed at the edge of networks [21]. Each Fog node is equipped with onboard computational resources, data storage, along with network communication facilities to bridges IoT and Cloud within the IoT network [22]. Moreover, Fog Computing acts as a bridge between the Cloud and Edge by enabling computing, storage, networking, and data management on network nodes within the proximity of IoT devices [23]. In recent years, Fog Computing has supported several applications such as smart home, smart grid, smart vehicle, health data management [24], and smart agriculture [25,26]. For example, smart home is a well-known application where Fog Computing offers more security, ultra-low latency, and efficient cost and energy [27]. The authors of [27] categorised Fog-based approaches in the smart homes as resource management (scheduling, allocation, provisioning, and load balancing) and service management (monitoring, security, energy management, and remote controlling). Moreover, Fog Computing provides vital support for processing large amounts of data produced and consumed by IoT sensors and devices, tractors, drones, applications, machines, and users. The main difference between Cloud Computing and Fog Computing is that the data can be accessed offline as some amount of data are stored in a local data centre in Fog Computing, but this is not true in Cloud. In comparison to Cloud, energy consumption and operational costs in the Fog Computing paradigm are low [28]. Moreover, the unique characteristics of Fog are low latency, real-time interaction, support for mobility, improvement of security, efficiency, and conserving network bandwidth [29,30]. These unique characteristics make agriculture easy for farmers and agricultural stakeholders. For example, the data collected by all devices may contain sensitive information, and they need to be processed quickly and locally. Therefore, Fog Computing can provide a benefit in such a way to do local processing and analysing without sending to the Cloud.

### 2.4. Edge Computing

Edge Computing is an emerging area where data processing occurs near proximity to mobile devices or sensors. As discussed above, Cloud Computing faces several severe issues due to centralisation. Therefore, Edge Computing has been proposed to improve the performance and overcome problems of Cloud by providing data processing and storage ability at the end devices locally. Ref. [31] stated that *Edge Computing refers to the enabling technologies allowing computation to be performed at the Edge of the network, on downstream data on behalf of Cloud services and upstream data on behalf of IoT services.* The distinguishing characteristics of Edge Computing from Cloud are dense geographical distribution, mobility support, location awareness, proximity, low latency, context-awareness, and heterogeneity [32]. Edge Computing is more or less the same as Fog Computing in terms of low latency, low bandwidth costs, mobility support, high scalability, and virtualisation service [33]. However, it has more limited resources, limited computation and storage capabilities, and proximity to end devices than Fog [33]. Mist computing is another type of computing that is the extreme edge of a network, typically consisting of micro-controllers and sensors. Mist computing uses microcomputers and microcontrollers to feed into Fog computing nodes [34]. Edge Computing is mainly contributing to agricultural applications such as pest identification, safety traceability of agricultural products, unmanned agricultural machinery, agricultural technology promotion, and intelligent management [35]. Moreover, Edge computing enables the evolution to 5G by bringing Cloud capabilities near to end-users [36,37,38]. However, it is essential to combine Edge with other computing such as Cloud and Fog to get the maximum benefits in agriculture.

### 2.5. Edge Computing and Fog Computing

The above two sections explain the details of Edge Computing and Fog Computing. As mentioned in the previous section, it is common in the literature to find that Edge and Fog Computing are defined as the same concept. The principal aim of these two concepts is to bring Cloud services and resources closer to the edge devices generating data. This section explains the view by other researchers, significant differences, and similarities from reviews on these two paradigms.

As described in a review by [15], the idea of Edge Computing first appeared in the literature in 2004–2005 with the concept of pushing the application logic and data to the edge of the network. However, as mentioned before, Fog Computing was first mentioned and defined in 2012 by Flavio Bonomi at CISCO. Some authors believe that Fog Computing is one of the classifications of Edge Computing [32,35]. For example, Ref. [32] mentioned Cloudlets, Mobile Edge Computing, and Fog Computing as classifications of Edge. Other authors consider Edge as another type of Fog Computing [39,40,41]. The Open Fog Consortium [42] revealed that Fog Computing is often erroneously called Edge Computing, but there are key differences. Fog works with the Cloud, whereas Edge is defined by the exclusion of Cloud.

In addition to computation, Fog also addresses networking, storage, control, and acceleration. Table 1 explores key differences between Fog and Edge.

## 3. Recent Reviews on Smart Agriculture

In the past few decades, research on smart agriculture and IoT has become more popular. While reviewing the existing research on Cloud, Fog, Edge, or IoT-based smart agriculture, a few papers were chosen as relevant to this study and analysed based on the requirement of this section. The research papers such as [10,43,44,45,46,47] are the latest review papers on smart agriculture/smart farming and IoT, and Ref. [35] is the latest review on Edge Computing in the agricultural Internet of Things. A significant number of publications and reviews published in 2020, in recent years, clearly show that researchers are very much interested in the field of smart agriculture where applying IoT and other new computing technologies such as Cloud, Fog, and Edge.

Talavera et al. [48] surveyed IoT applications for agro-industrial and environmental fields from the papers published between 2006 and 2016. The authors of this paper classified four application domains corresponding to monitoring, control, logistics, and prediction. Moreover, infrastructure and technology used by IoT solutions in agro-industrial and environmental fields were organised into seven groups such as sensing variables, actuator devices, power sources, communication technologies, Edge Computing technologies, storage strategies, and visualisation strategies. The authors explored the areas of communications, energy management, monitoring, and logistics for agro-industrial and environmental applications published between May 2016 and July 2017. They also considered limitations and challenges discussed and proposed an IoT architecture for agro-industrial and environmental applications.

Farooq et al. [43] conducted a comprehensive survey on the state-of-the-art for IoT in agriculture. In this paper, authors discussed significant components of IoT based smart farming along with relevant technologies (Cloud and Edge Computing, Big Data analytics and machine learning, communication networks and protocols, and robotics), the network architecture of IoT involving network architecture and layers, network topologies used, and devices and protocols used in agriculture IoT. Moreover, this survey analysed application domains, relevant smartphone, sensor-based applications, and security and privacy issues in IoT-based agriculture.

Farooq et al. [44] presented a systematic study on IoT agricultural applications, sensors/devices, communication protocols, and network types. The authors also discussed the main issues and challenges: security, cost, inadequate knowledge of technology, reliability, scalability, localisation, and interoperability in agriculture. Additionally, this survey presented country policies for IoT-based agriculture.

In 2020, Navarro et al. [10] conducted a review on IoT solutions for smart farming, identifying the primary devices, platforms, network protocols, processing data technologies, and the applicability of smart farming with IoT to agriculture. This review also shows an evolution in the way data are processed in recent years and the types of network connections such as wired networks which are used in indoor scenarios (e.g., greenhouse), and wireless networks, which are used both in indoor and outdoor scenarios (e.g., arable lands, orchards). To the author’s knowledge, agricultural applications of IoT solutions are categorised as chemical control, crop monitoring, disease prevention, irrigation control, soil management, supply chain traceability, and vehicles & machinery control. This article found that an increasing number of management systems use complementary technologies that rely on Cloud and big data computing to process large amounts of data.

Zhang et al. [35] presented the concepts related to Edge Computing and agricultural IoT and investigates the combination of Edge Computing and Artificial Intelligence, Blockchain and Virtual/Augmented reality technology in the application of pest identification and crop classification, agricultural product safety traceability, unmanned agricultural machinery, and agricultural product promotion. Furthermore, the authors of this paper conducted a review on the challenges of Edge Computing such as task allocation, data processing, privacy protection and security, and service stability in agriculture.

Glaroudis et al. [45] surveyed IoT messaging protocols that are regarded as major options for IoT applications in smart farming. The authors presented, based on the most recent literature, seven protocols (MQTT, CoAP, XMPP, AMQP, DDS, REST-HTTP, and WebSocket), analysed and compared with respect to their performance, and measured in terms of relevant key indicators (latency, energy and bandwidth requirements, throughput, reliability, and security). The authors concluded that, currently, the safest option seems to be the MQTT protocol, either when it is applied in end-to-end network architecture or when a gateway–server architecture is used to collect the measurements.

Boursianis et al. [46] performed a survey on IoT and UAV technology applied in agriculture and described the main principles of IoT technology, including intelligent sensors, IoT sensors types, networks and protocols used in agriculture, as well as IoT applications and solutions in smart farming. The article also presented the role of UAV technology in smart agriculture by analysing the applications of UAVs in various scenarios, including irrigation, fertilisation, use of pesticides, weed management, plant growth monitoring, crop disease management, and field-level phenotyping. In this review, the authors listed types of sensors that can be used to measure and calculate the parameters of an agricultural field. The types of sensors are soil water content sensor, soil moisture content sensor, soil electrical conductivity sensor, pH sensor, weed seeker sensor, temperature sensor, and wind speed sensor. Furthermore, the article depicted IoT technologies’ main applications and benefits in smart agriculture such as WSN, Cloud Computing, Big data analytics, Embedded systems, and communication protocols.

Ferrag et al. [47] surveyed research challenges of security and privacy issues in the field of green IoT-based agriculture. This paper provided a classification of threat models against green IoT-based agriculture into five categories: attacks against privacy, authentication, confidentiality, availability, and integrity properties. This article also provided an overview of a four-tier green IoT-based agriculture architecture with the components of the agricultural sensors layer, Fog layer, the core layer, and the Cloud layer.

In 2021, Friha et al. [49] presented a comprehensive survey on emerging technologies of IoT-based smart agriculture. In this paper, the authors provided a list of technologies such as UAV, wireless technologies, open-source platforms, SDN and NFV technologies, cloud and fog computing, and middleware platforms. Moreover, it is also analysed supply chain management solutions for agricultural IoTs based on blockchain. The papers also provided real-life smart farming projects that utilise emerging technologies and discussed the challenges such as hardware boards, interoperability of systems, networking and energy management, security and privacy threats, hardware and software costs, and education challenges.

Idoje et al. [50] discussed an overview of the various state of the art intelligent technologies on smart farming, crop, animal production, and post-harvesting. The authors also covered the impact of climate on smart farming. Islam et al. in [51] explored different use cases of smart farming, advantages, and applications of implementing IoT and UAVs in agriculture. The authors also discussed open research issues such as hardware maintenance and limited energy resources, security and privacy issues, big data in smart farming, weed detection and management, multi/hyper-spectral imagery for disease and pest control, and automated watering control and management in remote areas.

Although most of the reviews focused on general IoT applications in the agricultural field, which illustrated in Table 2, to our knowledge, no one has explicitly focused on smart agriculture and the combinations of Cloud, Fog, and Edge Computing. The applications are based on only Cloud, and applications based on combinations of computing (Cloud-Fog, Cloud-Edge, Cloud–Fog–Edge) can perform differently in terms of performance and cost. For example, Cloud-only-based innovative applications that deal with vast amounts of data must send or store in a central Cloud. This process is costly on both the edge and cloud side. Furthermore, the data sent to the Cloud may or may not be helpful or necessary. If the application is based on Fog-Cloud, then data processing can be done at the edge of the network, Fog, and only the crucial data can be sent to the Cloud. In this way, intelligent applications can reduce costs and provide real-time data processing with constant and high speed, low latency, and high internet bandwidth. Therefore, there is a need for a separated and specific review on the combinations of Cloud–Fog–Edge computing in the Smart Agricultural domain.

In this review, our primary goal is to identify the specific smart agricultural applications which have used Cloud, Fog, and Edge combinations. Our contributions in this work are: provide applications domains which are determined from existing Cloud–Fog–Edge applications, present research approaches for smart agriculture with Cloud, Fog, and Edge, emphasize Cloud–Fog–Edge applications for Smart Agriculture with detailed information of main contributions/achieved objects and components of the architecture, and a brief introduction of the proposed three-layered architecture for smart agriculture. Finally, we also discuss the challenges and future directions in Cloud–Fog–Edge based Smart agricultural applications.

## 4. Research Methodology

This research used the Preferred Reporting Items for Systematic Reviews and Meta-Analyses (PRISMA) [52] to conduct this systematic review. This research aims to investigate and provide a review of existing research on Cloud, Fog, and Edge Computing applications in the agricultural field.

Selected search engines and digital libraries were chosen based on their scientific contents and closely related to the objective of this paper. The chosen databases were ACM, IEEE Xplore, MDPI, ScienceDirect, and Springer which is illustrated in Table 3. A total of 2788 articles were obtained from 2015 to the present. From the 2788 resulting articles, 2706 were removed and those that did not contain the concepts searched for (smart agriculture or smart farm or precision farm or precision agriculture, and Cloud, Fog, Edge) in their title, as keywords or in summary, in any of their combinations. The abstracts of the remaining 82 were then analysed to see if they satisfied the research objectives and questions. This led to the further elimination of 27 articles, resulting in a total of 55 final articles that were analysed in depth. Thus, data have been obtained for 55 articles, which are analysed in the following sections. These data have been refined and clarified in subsequent stages. A flow diagram illustrating our systematic review of the whole process is shown in Figure 1.

### 4.1. Research Questions and Objectives

The research questions and the rationale of these questions are:**RQ1:** What key agricultural domains are covered? To identify in which agricultural domains the researchers’ focus contributes;**RQ2:** What research approaches are focused on existing works? To identify the research approaches;**RQ3:** What are the Cloud, Fog, and Edge applications for smart agriculture? To identify applications used in smart agriculture;**RQ4:** What components are used in the architecture? To create own architecture for future work;**RQ5:** What combinations of computing are used? To find in which combinations researchers have been interested in recent years;**RQ6:** What is the future direction of and what opportunities exist for smart agriculture? To identify and propose new solutions in the future.

### 4.2. Search String

A search string defined to gather published articles related to the research topics. The strings used in search are “Smart Farm”, “Precision Farm”, “Smart Agriculture”, “Precision Agriculture”, “Cloud”, “Fog”, and “Edge”. The publication date of the articles limited to years between 2015 and present. The query used for the database is ((“Smart Farm” OR “Precision Farm”) AND (Cloud OR Fog OR Edge)) OR ((“Smart Agriculture” OR “Precision Agriculture”) AND (Cloud OR Fog OR Edge)).

## 5. Discussion

A detailed discussion about application domains, research approaches, and architecture components of each layer, such as Edge, Fog, and Cloud has been presented in this section. Figure 2 shows the findings of this research from the surveyed papers. Finally, this section also briefly introduces the proposed architecture for future work from the observations gained from the reviews.

### 5.1. Application Domains

The agricultural applications domain can be mainly categorised as arable (crop) farming or pastoral (livestock) farming applications. The arable farming applications include crop management [53,54], soil management [55,56], irrigation management [57], weather management [58,59], and greenhouse management [60,61]. Moreover, pastoral or livestock farming includes animal monitoring [62,63,64] such as monitoring diseases and behaviour. However, researchers have categorised these domains from different perspectives. For instance, in 2017 [48], classified monitoring (air monitoring, soil monitoring, water monitoring, plant monitoring, and animal monitoring), control (irrigation control, fertiliser and pesticide control, illumination control, and access control), logistics (production, commerce, transport), and prediction (environmental conditions, production estimation, and crop growth) as main agricultural technological domains.

In 2019, Ref. [43] listed main domains of IoT agriculture applications as precision farming (climate conditions monitoring, soil patterns, pest and crop disease monitoring, irrigation monitoring system, determining the optimal time to plant and harvest, tracking and tracing, farm management system), livestock monitoring (animal health monitoring, heat stress level, physical gesture recognition, rumination, heart rate, GPS based monitoring), greenhouse monitoring (water management, plant monitoring, climate monitoring), and agricultural drones. Moreover, according to Ref. [65]’s view, smart agriculture domains are water management, animal care, crop care, farmer helpline, and farm management.

In 2020, Ref. [44] categorised the most common domains as monitoring, controlling, and tracking. At the same time, the authors identified soil monitoring, irrigation monitoring & controlling, humidity monitoring, temperature monitoring, air monitoring, precision farming, fertilisation monitoring, water monitoring and controlling, disease monitoring, and animal monitoring and tracking as agricultural application types. Ref. [10] identified that the most common applications of IoT solutions for smart farming are chemical control, crop monitoring, disease prevention, irrigation control, soil management, supply chain traceability, and vehicles and machinery control.

Based on the existing categories, it is clear that no one has followed any methods or strategies for the agricultural domain groupings. Therefore, to answer the first research question (RQ1), we decided to categorise according to the principle scope of smart agriculture. The results obtained in the analysis from the selected studies were grouped to identify the main types of Cloud, Fog, and Edge Computing agricultural applications. The identified types are (1) Animal Management, (2) Crop Management, (3) Greenhouse Management, (4) Irrigation Management, (5) Soil Management, and (6) Weather Management. Results are summarised in Table 4 and Figure 3.
Animal Management: Animal management or livestock management involves all the activities such as animal health and welfare, feeding, grazing and pastoralism, breeding, and animal husbandry carried out by farmers to raise farm animals. Livestock management plays a crucial role as part of human life. Therefore, the demand for high-quality dairy products increasing day by day and precision livestock is also considering as a significant category in smart agriculture by researchers. For instance, Refs. [62,63] implemented a system for animal behaviour analysis and health monitoring in a dairy farming scenario. In order to monitor animal welfare, the authors of [64] developed an open-source system. Additionally, as part of multi domain systems, Refs. [59,66] have also contributed to animal management.Crop Management: Crop management includes all activities used to improve the growth, development, and yield of crops. Ref. [53] proposes the use of smart drones to manage crops in terms of (a)identify pests, weeds and diseases which help in optimizing pesticide usage and crop sprays, (b) estimate the crop yield, (c) provide data on soil fertility by detecting nutrient deficiencies, and (d) measure irrigation and control crops by identifying areas where water stress is suspected. The authors in [54] proposed a smart model for the agriculture field to predict the crop yield and decide a better crop sequence. Ref. [67] developed a smart robotic system to improve harvesting and production, and Ref. [68] developed an application to identify real-time pest detection. Moreover, other applications identified [25,58,59,66,69,70,71] as multi-domain applications.Greenhouse Management: There were few applications proposed and developed for greenhouse management in terms of the home automation system to control environmental conditions [61], flexible platform able to cope with soilless culture needs in full re-circulation using moderately saline water [60], and a vegetable growing cabinet [72]. In addition, Ref. [73] implemented an agricultural data collection framework and experimented in a greenhouse in order to analyse the proposed methods. Ref. [74] implemented a wireless agricultural monitoring system for greenhouse.Irrigation Management: Irrigation is crucial for all activities in both animal and crop farming. Good irrigation scheduling and efficient utilisation of water resources are two main parameters in agricultural systems. Therefore, in order to get the maximum utilisation, Ref. [75] proposed architecture and developed a system for intelligent irrigation monitoring. Similarly, Ref. [76] presented a low-cost solution for automatic Cloud-based irrigation, and Ref. [77] implemented a Cloud and IoT based system for irrigation schedule. Moreover, Ref. [78] implemented a smart irrigation systems. See Table 4 for other multi-domain applications.Soil Management: Every plant needs certain moisture levels for its optimal growth to be maintained [55]. The soil management refers to all the processes which aim to improve soil performance. In order to achieve this target, the authors of [55] developed a low cost, continuous monitoring of the soil moisture system. In addition, Ref. [56] proposed a model for efficient soil moisture monitoring. Other applications such as [53,57,58,71,79,80,81] are pointed out as multi domain applications.Weather Management: The authors of [82] proposed a remote farm monitoring system to monitor temperature, humidity, and soil moisture. In addition, Ref. [83] presented an architecture to monitor environmental data such as wind speed and direction, rain volume, and air temperature and humidity. Ref. [80] proposed a multi-domain system that also covers environmental monitoring parameters like temperature, air humidity, and rainfall. The authors in [84] developed an Agricultural Environment Management System (AEMS) to monitor temperature and humidity. In contrast, Ref. [58] implemented a prototype for the monitoring and predicting soil moisture, humidity, light, and temperature data. Ref. [85] developed a system for environmental monitoring in olive groves. Furthermore, Refs. [58,59,70,80,81] identified multi-domain categories where Ref. [58] developed a system to monitor and predict the data of soil moisture, humidity, light, and temperature. Ref. [81] proposed a home automation system to monitor temperature and humidity. Finally, Ref. [86] proposed an architecture to monitor temperature, humidity, light intensity, and soil moisture in a coffee farm.

Figure 3 illustrates the number of single and multi-domains research papers published in the agricultural application domains between the year 2015 and the present. It can be seen that crop management has more publications than other domains, where most of them are from the multi-domain category. In terms of single domain applications, greenhouse management has more than multi-domain applications. However, crop, irrigation, soil, and weather management applications are more from multi-domain types. Overall, we can see that most of the agricultural applications are developed not only for a single domain but also combined with other applications.

### 5.2. Research Approaches

The ultimate goal of this section is to find the solution for RQ2. In this section, we discuss the types of research approaches from the analysed research papers in agricultural Cloud, Fog, and Edge Computing applications. They have presented a particular innovation and output under Survey/Review, Architecture, and System/Application. Each of them is pointed out briefly to determine a common concept according to terminology as follows:Survey/Review: A process of analyzing, summarizing, organizing, and presenting novel conclusions from the results of technical review of recently published scholarly articles. It is mainly a comprehensive review of Cloud, Fog, Edge, and IoT based smart agriculture.System/Application: It is defined as a software program or group of programs designed for end-users as a solution for a specific problem.Architecture: It is a general abstract design of an application or system that tries to satisfy the business needs according to requirements, limitations, and technical constraints. It contains the components, functions, and communications. Furthermore, it focuses on how they interact with each other components and with users.

In Table 5, we summarise the identified research approaches. From the analysed recent studies, few papers [35,91,92,93] were identified under the Survey/Review category. For instance, Ref. [91] reviewed the main trends and challenges in smart climate agriculture. The authors of [92] surveyed the state-of-the-art research utilising the Edge model of computing in agriculture, whereas Ref. [93] studied typical applications of agriculture IoT Sensor monitoring network technologies using Cloud Computing. Ref. [94] reviewed Wearable Internet of Things (WIoT) enabled precision livestock farming in smart farms.

The applications and systems about smart agriculture and modern computing paradigms were observed from our survey. The notable applications are Decision Support System (DSS) for potato late blight disease prevention [69], Cloud of Things (CoT)-based automated irrigation system [75], Cloud-based digital farm management system for vegetable production process management and quality traceability [96], Fog assisted application for animal behaviour analysis and health monitoring in dairy farming [62,63], a smart agricultural knowledge support system to provide real-time information [80], smart farm computing systems for animal welfare monitoring [64], and a greenhouse system [61].

Moreover, some studies presented architecture models for the smart agricultural domain. For example, Ref. [86] introduced a Fog-Cloud based architecture approach to provide data collection reliability in Things-based Coffee Smart Farming. Ref. [58] proposed architecture for monitoring and predicting data in precision agriculture. Ref. [97] presented an architecture based on Fog Nodes and LoRa technology to optimize the number of nodes deployed in smart farms. Ref. [83] proposed three-layer architecture that consists of a front-end layer, gateway layer, and back-end layer for farm monitoring. Ref. [56] introduced a Cloud-based architecture for soil moisture monitoring. Ref. [84] proposed an integrated WSN and Cloud architecture for agricultural environment applications such as temperature, humidity, moisture, and pH. Ref. [79] proposed Home Edge Computing architecture (HEC) for smart and sustainable agriculture and breeding. It is worth mentioning that, in all of the proposed and implemented architecture, the bottom layer is always sensors, whereas the top layer is Cloud centre or Cloud server. The middle layer was different based on the author’s preferred combination, Edge, Fog, or Edge-Fog.

Finally, it is observed that some authors not only proposed an architecture but also implemented systems as well. More specifically, Ref. [59] proposed architecture of a Cloud-based autonomic information system for agriculture to manage different types of agriculture-related data. The authors also developed a system to get information from IoT devices, analyze them, and store user data in the Cloud. Similarly, Ref. [60] presented three-tier architecture using Cloud, Edge and Cyber-Physical Systems (CPS) planes to manage greenhouse facilities, and Ref. [25] proposed and developed a platform by integrating Cloud with limited network information resources to be integrated and automated, including agricultural monitoring automation, pest management image analysis and monitoring. Ref. [66] implemented an architecture using the Industry 4.0-oriented Edge Computing reference architecture (GECA) and developed an application for monitoring and managing mixed crop-livestock farming. Ref. [85] proposed three-layered Cloud/Fog Computing network architecture and developed a system for monitoring olive groves. Furthermore, Refs. [62,81,84] also proposed an architecture and implemented a system.

### 5.3. Existing Applications on Cloud, Fog, and Edge Computing and Smart Agriculture

Table 6 presents a summary in order to answer the research questions (RQ3, RQ4 and RQ5) based on the information extracted from the selected studies. First of all, we have grouped the components of architecture into Edge, Fog, and Cloud layers according to the existing models.
Edge layer: We observed that, in most of the applications, sensors [55,58,61,63,64,67,69,79,81,98], actuators [67,82,85,97], or IoT devices [60] were used as a bottom layer. The most common sensors used in the applications are wearable sensors, environmental sensors such as temperature, humidity, light, soil moisture, pH, and satellite sensors [56]. In particular, Ref. [66] considered barn sensors, agro-meteo station in crops, and cattle sensors as the IoT layer, and the authors of [86] included sensing systems such as temperature, humidity, light intensity, and soil moisture. Ref. [66] used local data store and Edge gateways in the Edge layer. Ref. [73] considered Edge servers in the Edge layer. In some papers, authors used different components in the Edge layer. For instance, Edge Gateways [82], gateway such as WiFi [85], Edge node, communication interfaces, gateway [80], MEC and HEC [79], and NFV nodes [60]. To make an interaction between the Edge layer and Cloud, wireless technologies such as LoRa, Wi-Fi, 3G, or ZigBee can be used.Fog layer: Ref. [86] proposed Fog hierarchical architecture where authors introduced two Fog layers with the components of Fog controllers in the first layer and Fog nodes as the second. Ref. [85] presented a Fog Computing network as a Fog layer where it contains storage, server, and network attached storage. In some papers, authors used different components in the Fog layer. For instance, Fog gateways [82], Fog nodes [63,97], farm controller [64], Fog node, and gateway [60,80,98].Cloud layer: Ref. [66] considered Cloud applications and APIs as a Cloud layer, whereas Ref. [86] presented Data Centers, SaaS, PaaS, and IaaS in the Cloud layer. Refs. [53,73] have a Cloud center in the Cloud layer. In some other papers, authors used different components in the Cloud layer. For instance, Cloud Servers and its service [82], Cloud (data storage, data analytics, data visualization, APIs) [85], Data processing center [56,60,67,75], Cloud database [63], Cloud services (Cloud server and KB) [80], Cloud application [64,81], central Cloud [79], Cloud server [58,98], Cloud API [55], and Cloud KB and Resource pool [59],

We have noticed that the different combinations have been used in the surveyed applications—for example, Edge-Cloud, Fog-Cloud, Edge–Fog–Cloud, and Cloud computing applications. As shown in Table 6, the most common combination of applications are from the Fog-Cloud [25,60,62,63,64,75,85,86,97,98]. Moreover, most of the applications were Cloud-based applications with IoT sensors as the bottom layer [53,55,56,57,58,59,61,69,70,71,74,76,78,81,83,84,87,89]. Similarly, few applications [80,82] used the Edge–Fog–Cloud combination while Refs. [66,73,79] applied the Edge–Cloud combination. In addition, from Table 6, we can observe that all applications have used sensors, actuators, IoT devices, or local gateways as IoT layers to collect the data from the agricultural field.

Table 6 also reveals the information of achieved benefits of the applications. The majority of the papers developed the applications with a low cost requirement [25,53,55,56,64,67,69,74,75,76,80,81,85,89]. As all these applications are built in smart farms, we can consider this as a valuable feature. Moreover, the use of combinations also brings the benefits such as low latency [59,79,80,97] and saving on bandwidth [59,97]. Additionally, the applications solved the problems when they deal with data in terms of reliable data collection [86], reduction of data redundancy [73], data storage [53], data security [80,97], real time analytics [62,63], reduction in data traffic [66,98], and maximum utilisation of data [84]. Furthermore, in some papers, authors have mentioned that efficiency [56,60,69,75,80,83] is one of the major advantages, and some other authors pointed out more specifically what is efficient such as resource utilisation [53,97,98], power utilisation [82,85], low execution time [59], efficient decision-making [57,58], and automation [57,59,60,76,81].

Furthermore, different types of wireless technologies are used for communication purposes in each layer. The most common used technologies in the surveyed papers are Zigbee [61,69,81,85], LoRa [69,79,82,97], Bluetooth [69], Near field communication (NFC), Wi-Fi [66], and nRF [82]. For each level layer, different protocols are used by authors. For example, Ref. [82] used nRF wireless to send the data from sensors to Edge nodes. The authors also used LoRa to send the processed and compressed data from edge nodes to fog gateways and used Wi-Fi, Ethernet, or 4G for Fog gateways to Cloud interaction. To make communication between sensors and nearest gateways, some authors used nRF [82,83] wireless communication module.

### 5.4. Proposed Architecture

Based on the observations gained through the reviews, a Cloud–Fog–Edge Computing model proposed for smart farming is illustrated in Figure 4. The Cloud layer is mainly for ample scale data storage and data analytics. This layer is also responsible for loading algorithms and data analytical tools to Fog nodes. This can also be used to store backup data for future analysis. The Fog layer is essential in this model, and this will be installed in local farms. Fog layers will be responsible for real-time data analytics such as predicting pests and diseases, yield prediction, weather prediction, and agricultural monitoring automation. Moreover, this will make decisions on real-time data and do reasoning analysis as well. Finally, the processed and analysed data can be uploaded to the Cloud layer for backup purposes or further analysis. The third layer is the Edge, consisting of end devices, tractors, sensors, and actuators. The main goal of this layer is the collection of data and its transfer to the Fog layer.

## 6. Challenges and Future Directions

Although Cloud, Fog, and Edge-based technologies within the agricultural sector present several advantages, it has some notable challenges as well. This section discusses the challenges and future directions in agricultural-based Cloud, Fog, and Edge-based computing which aims to answer RQ6. Security and privacy [35,91,99,100,101,102] require mobility support and widespread geographical distribution through an extended number of nodes [91], constant and high speed network connectivity [91], strong need of real-time data processing [35,91,103,104], better power management [35,103,105], high hardware costs [35,104,106], and poor internet connectivity in farms [7,100,107] are few challenges. The following clearly describe the challenges and possible solutions for them.
Security and privacy [35,91,99,100,101,102]: When agricultural applications deal with Cloud computing, data security and privacy, authorisation and trust, authentication and secure communication, and compliance and regulations are the significant challenges [108]. This is because, in smart farms, an enormous amount of data are generated from various kinds of data resources such as sensors, actuators, and Edge devices. Therefore, for the data stored in the Cloud, there is a chance of leakage. This might cause severe economic loss for farmers and agricultural industries. However, to overcome this issue, applications must include more computational capabilities, such as edge computing, handling massive data, artificial intelligence resources, and security features, with the combination of Cloud [101].Requirement of mobility support [91]: Smart farms require mobility support and real-time data processing since it continuously collects more data from the field. If the farms are only connected with Cloud, these features are not possible. However, the characteristics of Fog make it possible to do real-time data collection and processing at the farm. Additionally, for real-time data processing, a consistently high speed is essential. To solve this issue, the combinations of Fog and Edge are recommended because these two have characteristics such as low latency, high bandwidth, and high mobility [91].Data processing [35,91,103,104]: Data processing and decision-making are crucial features in smart agriculture. If smart farms only depend on Cloud to analyse and produce the results, it will not be a good solution in terms of real-time data. In this case, the combination of Edge–Cloud or Fog–Cloud would be an excellent solution.Better power management [35,103,105,109]: Smart farms are not possible without sensors, actuators, and mobile devices. All of these devices depend on power to collect data and send it to other layers or processing in the edge node. Efficient energy and power management strategies enhance the lifetime of batteries [110,111,112]. As an alternative, renewable energy sources such as solar power can also be used for the longer life of sensor nodes [103,109,113].High hardware costs [35,104,106]: As sensors and other Edge devices continuously collect data and send it to the Cloud in Cloud-based agricultural applications, the process of uploading and analysing data will consume not only hardware resources but also a lot of network resources and Cloud resources. Additionally, it also includes the deployment of IoT in smart farms. Efficient cost management is highly needed to manage hardware cost issues in smart agriculture.Poor internet connectivity [7,100,107]: This is one of the most common issues in smart farms, especially in rural areas. Internet connectivity is the essential thing to be a smart farm. However, poor internet connectivity in farms causes some issues such as data loss, processing delay, slow data upload speed, and slow response. Moreover, these problems will happen if the smart farm is only connected with Cloud. However, Fog Computing provides the facility to solve these issues since it has its own local server and data centre. Therefore, local data processing is possible as it has offline services as well.

All the above identified challenges are mainly from Cloud based applications. The combinations of computing and Fog based applications can solve most of the problems as we discussed above. From Table 6, we can observe that the combinations of modern paradigms in the agricultural based applications proved the high possibility to solve the common problems of Cloud such as latency [59,79,80,97], bandwidth [59,97], and network traffic issues [66,98]. It is also proved that most of the applications are low-cost applications. Moreover, it is worth mentioning that Fog Computing can solve some major problems such as real-time data processing with low latency and high bandwidth, unnecessary cost, and data security and privacy. However, further research needs to be done to find out how to overcome these issues.

## 7. Conclusions

Fog and Edge Computing are emerging paradigms compared to the Cloud. In recent years, researchers focused more on combinations of these new paradigms to build systems, including the agricultural domain. In this study, we have presented an updated systematic literature review on the role of modern computing paradigms, namely Cloud, Fog, and Edge Computing in Smart agriculture domains in terms of application domains, research approaches, and existing applications with the combination. Based on an analysis of the literature, the agricultural application domains were identified as six categories: animal management, crop management, greenhouse management, irrigation management, soil management, and weather management. This work also discussed the identified benefits and objectives of the surveyed smart agricultural applications (i.e., low latency, low cost, saving bandwidth, and reducing data traffic). Additionally, it reported the methodology and intermediate results obtained during the stages of identification, screening, and results in great detail. From 2788 initial studies extracted from electronic sources, 55 primary studies were selected based on their relevance to answering research questions.

Furthermore, it was observed in this review that, in recent works, most of the research on smart agricultural applications is mainly focused on Cloud-based and sensors. Few scholars have applied combinations of Cloud, Fog, and Edge in the agricultural domain. It also found that most applications relied heavily on Cloud, and few applications started to apply the combinations of computing paradigms such as Cloud-Edge and Cloud-Fog. This review also recognized few challenges in existing smart agriculture based applications such as security and privacy, the requirement of mobility support, data processing, better power management, high hardware costs, and poor internet connectivity. It is also reasonable to assume that future solutions will need to highly focus on applying the combinations in order to get the benefits of a truly connected and smart farming concept. In the next step, we will focus on implementing the proposed architecture model with the combinations of all three computing for smart farms in order to get maximum benefits of each paradigm. According to the GSM Association, the number of devices is expected to increase in the following years. Therefore, we firmly believe that combining this architectural implementation will bring more advantages and new opportunities in the smart agricultural domain.

## Figures and Tables

**Figure 1 sensors-21-05922-f001:**
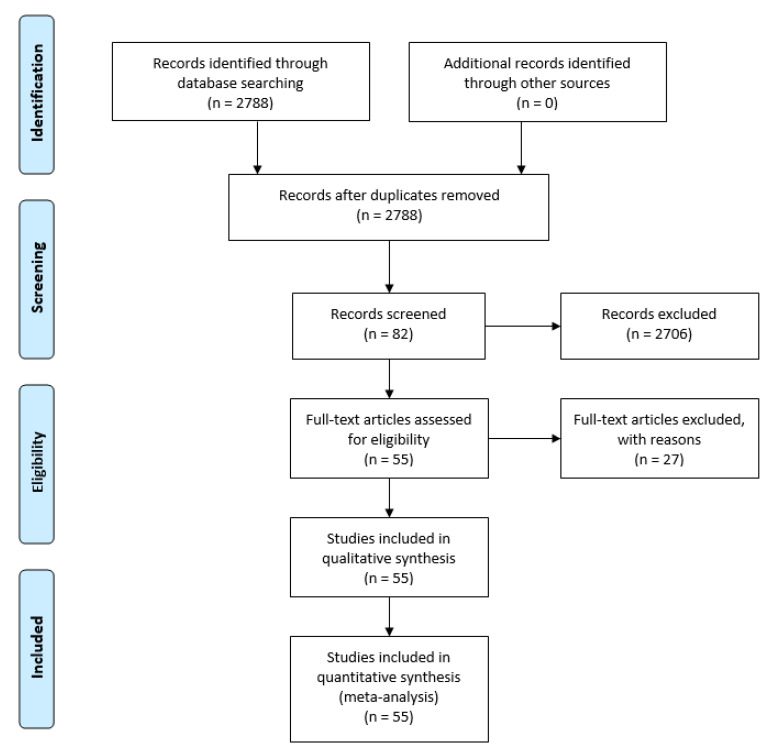
Flow diagram to show the study-selection process.

**Figure 2 sensors-21-05922-f002:**
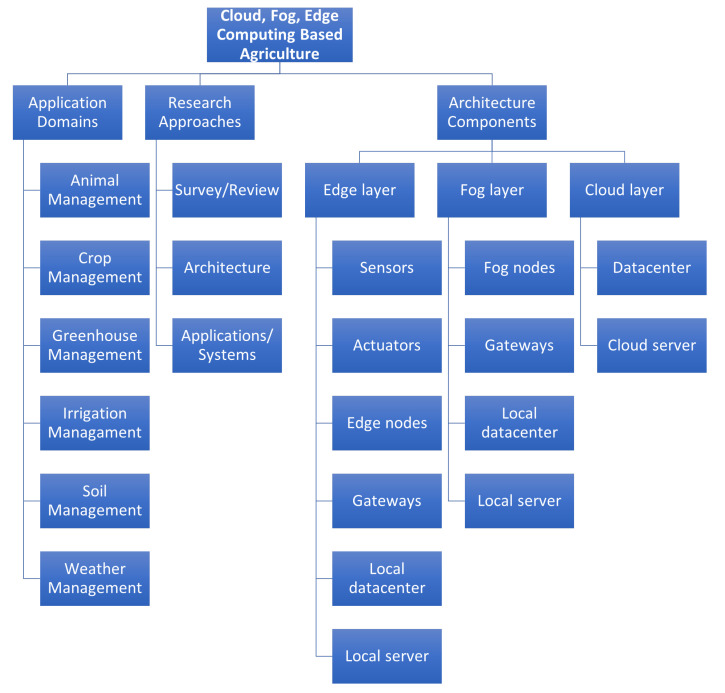
Hierarchy.

**Figure 3 sensors-21-05922-f003:**
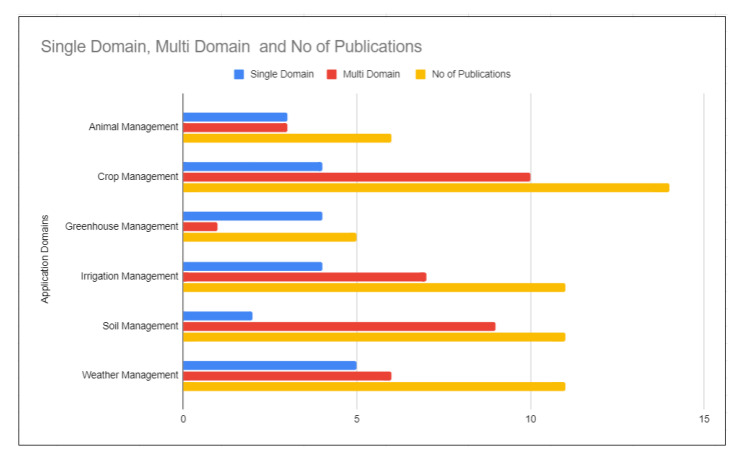
Application domains.

**Figure 4 sensors-21-05922-f004:**
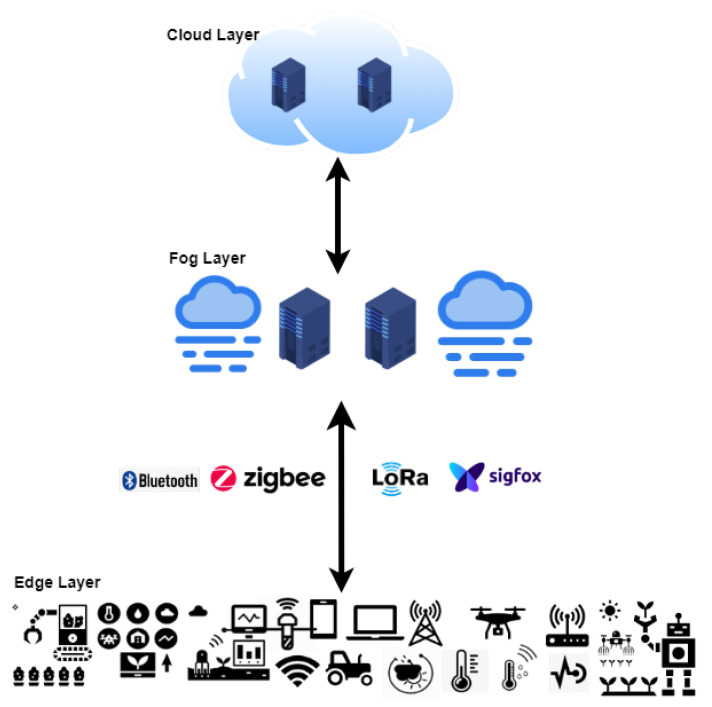
Three layered architecture for smart agriculture.

**Table 1 sensors-21-05922-t001:** Comparison of Fog and Edge.

Features	Edge Computing	Fog Computing
Location of data collection, processing, storage	Network Edge, Edge devices	Near Edge, Core networking
Computation and storage capabilities	More limited	Limited
Resources	More limited	Limited
Handling multiple IoT application	Unsupported	Supported
Focus	IoT level	Infrastructure level

**Table 2 sensors-21-05922-t002:** Review/Survey Papers and their Contribution.

Year	Reference	Title	Main Focus/Contribution
2017	[48]	Review of IoT applications in agro-industrial and environmental fields	A review on agro-industrial and IoT environmental applications and identified application areas, trends, architectures, and open challenges from the papers published from 2006 to 2016.
2019	[43]	A Survey on the Role of IoT in Agriculture for the Implementation of Smart Farming	A comprehensive survey on the state-of-the-art for IoT in agriculture and discussed agricultural network architecture, platform, and topology.
2020	[44]	Role of IoT Technology in Agriculture: A Systematic Literature Review	Presented a systematic literature review on collection of all relevant research on IoT agricultural applications, sensors/devices, communication protocols, and network types from selective high-quality research articles published in the domain of IoT-based agriculture between 2006 and 2019.
2020	[10]	A Systematic Review of IoT Solutions for Smart Farming	Presented a systematic review to identify the main devices, platforms, network protocols, processing data technologies and the applicability of smart farming with IoT to agriculture.
2020	[35]	Overview of Edge Computing in the Agricultural Internet of Things: Key Technologies, Applications, Challenges	Review on application of Edge Computing in the Agricultural Internet of Things and investigates the combination of Edge Computing and Artificial Intelligence, blockchain, and Virtual/Augmented reality technology.
2020	[45]	Survey, comparison, and research challenges of IoT application protocols for smart farming	A survey of research efforts on the IoT application layer protocols, focusing on their basic characteristics, their performance, as well as their recent use in agricultural applications.
2020	[46]	Internet of Things (IoT) and Agricultural Unmanned Aerial Vehicles (UAVs) in smart farming: A comprehensive review	A survey on main principles of IoT technology, including intelligent sensors, IoT sensor types, networks, and protocols used in agriculture, as well as IoT applications and solutions in smart farming.
2020	[47]	Security and Privacy for Green IoT-Based Agriculture: Review, Blockchain Solutions, and Challenges	Presented research challenges on security and privacy issues in the field of green IoT-based agriculture.
2021	[49]	Internet of Things for the Future of Smart Agriculture: A Comprehensive Survey of Emerging Technologies	Comprehensive survey of emerging technologies for IoT based smart agriculture.
2021	[50]	Survey for smart farming technologies: Challenges and issues	An extensive review of the use of smart technologies in agriculture and elaborates the technologies for smart agriculture including, Internet of Things, cloud computing, machine learning, and artificial intelligence.
2021	[51]	A Review of Applications and Communication Technologies for Internet of Things (IoT) and Unmanned Aerial Vehicle (UAV) Based Sustainable Smart Farming	Reviewed some major applications of IoT and UAV in smart farming, and explored the communication technologies, network functionalities, and connectivity requirements for Smart farming.
2021	This Survey	A Systematic Survey on the Role of Cloud, Fog, and Edge Computing Combination in Smart Agriculture	A systematic Survey on Cloud, Fog, and Edge Computing applications, architecture components from research articles published between 2015 and up-to-date (2021-June) from the domain of Cloud, Fog, and Edge based agriculture.

**Table 3 sensors-21-05922-t003:** Databases used in the search phase.

Database	URL
ACM Digital Library	https://dl.acm.org/ (accessed on 30 June 2021)
IEEE Xplore	https://ieeexplore.ieee.org/Xplore/home.jsp (accessed on 30 June 2021)
MDPI	https://www.mdpi.com/ (accessed on 30 June 2021)
Science Direct	https://www.sciencedirect.com/ (accessed on 30 June 2021)
Springer	https://link.springer.com/ (accessed on 30 June 2021)

**Table 4 sensors-21-05922-t004:** Application domains.

Application Domains	Single Domain Applications	Multi Domain Applications	No. of Applications
Animal Management	[62,63,64]	[59,66,87]	06
Crop Management	[53,54,67,68]	[25,58,59,66,69,70,71,87,88,89]	14
Greenhouse Management	[60,61,72,74]	[73]	05
Irrigation Management	[75,76,77,78]	[25,53,57,59,79,88,90]	11
Soil Management	[55,56]	[25,53,57,58,59,71,79,80,81]	11
Weather Management	[82,83,84,85,86]	[58,59,70,80,81,89]	11

**Table 5 sensors-21-05922-t005:** Research approaches.

Research Approaches	Single Approach	Multiple Approach
Survey/Review	[35,49,91,92,93,95]	
Applications/Systems	[61,63,64,67,68,69,71,72,74,75,78,80,89,96]	[25,59,60,62,66,77,81,84,85]
Architecture	[53,56,58,73,76,79,83,86,87,97]	[25,59,60,62,66,77,81,84,85]

**Table 6 sensors-21-05922-t006:** Existing works.

Year	Reference	Description	Main Contribution or Achieved Objectives	Architecture Components		
				Edge Layer	Fog Layer	Cloud Layer
2021	[87]	Proposed a Climate-Smart architecture for fostering and supporting integrated agricultural systems.	automation	sensors, drones	-	Cloud
2021	[89]	Implemented a system for environmental smart farming monitoring systems based on IoT and UAVs.	low cost	sensors, drones	-	Cloud
2021	[78]	Provided architectural design and implementation of a smart irrigation system that uses a WSN based on Arduino and XBee technologies.	automation, efficient	sensors	-	Cloud
2021	[71]	Designed and implemented an online monitoring system for crop planting and soil remediation.	reliable and convenient data sources	sensors	-	Cloud
2021	[70]	Proposed an IoT-based Smart Agriculture System assisting farmers.	increase overall yield, increase quality of products	sensors	-	Cloud
2021	[74]	Proposed and implemented wireless agricultural monitoring system for greenhouse.	low cost	sensors	-	Cloud
2021	[77]	Implemented a cloud and IoT based system to automate the irrigation schedule.	automation	sensors	-	Cloud
2021	[68]	Developed an application to provide real-time pest detection in the orchard.	increase crop yield	drones	-	Cloud
2021	[67]	Developed an IoT based smart robotic system to improve harvesting and production.	low cost, consumes less power	sensors, actuators	-	Cloud (data storage)
2020	[53]	Introduction of a Smart Drone for crop management where the real-time drone data coupled with IoT and Cloud Computing.	promote resource sharing, cost-saving and data storage	sensors	-	Cloud (Server, Storage)
2020	[66]	Proposed an architecture and developed a system to monitor the state of dairy cattle and feed grain in real time.	reduction in data traffic, improvement in the reliability in communications.	sensors, IoT nodes, Edge Gateway, Local data store	-	Cloud Applications, APIs
2020	[98]	Proposed a strategy that assigns DL layers to Fog nodes in a Fog-computing-based smart agriculture environment	efficient resource utilisation, reducing network congestion	sensors	Fog nodes	Cloud server
2020	[86]	An approach that introduces a Things-Fog-Cloud architecture that combines ML and Interpolation techniques to intelligently and automatically provide data reliability on SF applications	reliable data collection	sensors	Fog controllers, Fog Nodes	Data Centers, SaaS, PaaS, IaaS
2020	[73]	Proposed an approach for data collection and experimented in a smart greenhouse.	reduce data redundancy	sensors, Edge server	-	Cloud center
2020	[85]	Proposed Latency Adjustable Cloud/Fog Computing Architecture for monitoring Olive groves.	low-cost, power-efficient	sensors	Fog (local storage, local server)	Cloud (server, storage)
2020	[80]	Proposed smart agricultural knowledge support system to provide real time information	efficiency, latency, cost level, scalability, speed, data security	sensors, Edge node, gateway	Fog node, gateway	Cloud services (Cloud server, KB)
2020	[58]	Proposed an architecture for the monitoring and predicting of data in precision agriculture.	sufficient decision-making	sensors	-	Cloud servers
2020	[97]	Proposed an architecture based on Fog Nodes and LoRa technology to optimize the number of nodes deployment in smart farms.	low latency, save bandwidth, low energy consumption, Data security	sensors, actuators	Fog node	Cloud
2020	[55]	Developed a soil moisture system with IoT and Cloud.	low cost, continuous monitoring	sensor	-	Cloud
2019	[79]	Proposed a Home Edge Computing Architecture (HECA) and implemented use cases for smart agriculture	low latency	sensors, Home Edge Computing (local data center, gateway), Mobile Edge Computing (gateway, data center)	-	Cloud
2019	[59]	Proposed an architecture and developed an autonomic system for delivering agriculture as a service.	high network bandwidth, low execution cost, low execution time, low latency, automation	sensors	-	Cloud
2019	[60]	Proposed and developed a flexible platform able to cope with soilless culture needs in full recirculation greenhouse.	efficiency in water consumption, automation	sensors, CPS, NFV nodes	-	Data Cloud
2019	[76]	Proposed and developed a design of monitoring and operating irrigation networks.	low-cost, automatic, high performance	sensors, gateways	-	Cloud
2019	[82]	Presented a hybrid 5-layer architecture for IoT systems in smart farms.	low power and long-range transmission	sensor nodes, actuator nodes, Edge gateways	Fog gateways	databases, application servers
2018	[62,63]	Proposed a Fog Computing based application for animal behaviour analysis and health monitoring in dairy farming.	efficient real time data analytics, affordable, scalable	sensors	Fog node	Cloud (database)
2018	[25]	Proposed a platform through Cloud integration for large-area data collection and analysis.	reduced cost of network transmission	IoT devices	Fog	Cloud
2018	[69]	Developed a Decision Support System for Late Blight disease.	efficient, minimal cost	sensors, local gateway	-	Cloud
2017	[75]	Proposed and developed CoT-based irrigation system.	improve irrigation efficiency, lower costs	sensors	Fog nodes, gateway	Data center
2017	[84]	Proposed an integrated WSN and Cloud architecture for agricultural environment applications.	fully utilise the data	sensors	-	Cloud (gateway, database)
2017	[64]	Proposed open and low-cost concepts for Fog Computing system to create a smart farm animal welfare monitoring system.	low cost	sensors	farm controller	Cloud applications
2017	[81]	Proposed a agriculture monitoring systems based on IoT with Cloud.	low cost, automation	sensors	-	Cloud applications
2016	[61]	Proposed greenhouse by using IoT and Cloud.	higher crop yield, better quality	sensors	-	Cloud
2016	[83]	Proposed a Cloud-based three-layer architecture for IoT precision agricultural applications.	efficient	sensors and actuators, gateway (WiFi)	-	Cloud (data storage, data analytics, data visualization, APIs)
2016	[57]	Proposed an extensible Cloud-based software platform for Precision Agriculture	decision support, automation	sensors	-	Cloud
2016	[56]	Proposed a Cloud Computing enables infrastructure for efficient soil moisture monitoring.	efficient, reduce costs	sensors	-	Cloud (data processing center)

## Data Availability

No new data were created or analyzed in this study. Data sharing is not applicable to this article.

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
