# Peer review of "A Systematic Survey on the Role of Cloud, Fog, and Edge Computing Combination in Smart Agriculture"

_sensors, 2021, doi:10.3390/s21175922_

Round 1

Reviewer 1 Report

I am pleased to send you the revision of the manuscript " A Systematic Survey on the Role of Cloud, Fog, and Edge Computing Combination in Smart Agriculture ". About this study, in my opinion, I believe that the review is very interesting and essential to set up and enhance the effective management of agricultural sector.
The manuscript is good structured, logically organized. The discussion presentation was efficient and appropriate.
The topic is perfectly fitting the scope of the Journal, and for this reviewer the manuscript contains valuable research information deserving publication

Author Response

Comments: 

I am pleased to send you the revision of the manuscript " A Systematic Survey on the Role of Cloud, Fog, and Edge Computing Combination in Smart Agriculture ". About this study, in my opinion, I believe that the review is very interesting and essential to set up and enhance the effective management of agricultural sector.
The manuscript is good structured, logically organized. The discussion presentation was efficient and appropriate.
The topic is perfectly fitting the scope of the Journal, and for this reviewer the manuscript contains valuable research information deserving publication

Response:

Thank you for your comments.

Reviewer 2 Report

The paper presents an interesting study on how cloud, fog and edge can assist in the area of sensor and sensor management.

The section one presenting current approach in edge and fog needs enhancement in order to include other paradigms like Mist computing or extreme edge concepts.

Further reference on the area can be found in the following reference DOI: 10.1109/MCOM.2017.1600730.

Another approach that recent is taking a lot of credits is the 5G use in the edge computing and work on the area can be found in the following reference DOI: 10.1109/MCOM.2017.1700105

The Figure 1 is not needed as it is explained inside the introduction section.

It will be interesting to see a Architecture comparison in the area and added in the Paper.

Author Response

Comments

  1. The paper presents an interesting study on how cloud, fog and edge can assist in the area of sensor and sensor management.
    Response: Thank you for your comments

  2. The section one presenting current approach in edge and fog needs enhancement in order to include other paradigms like Mist computing or extreme edge concepts.
    Response: We have added a brief introduction about Mist computing under the section Edge Computing 2.4

  3. Further reference on the area can be found in the following reference DOI: 10.1109/MCOM.2017.1600730.
    Response: We referred to this.

  4. Another approach that recent is taking a lot of credits is the 5G use in the edge computing and work on the area can be found in the following reference DOI: 10.1109/MCOM.2017.1700105
    Response: We have added some briefs on 5G under section 2.4

  5. The Figure 1 is not needed as it is explained inside the introduction section.
    Response: We have deleted Figure 1

  6. It will be interesting to see a Architecture comparison in the area and added in the Paper.
    Response: We also agree that Architecture comparison will be interesting. However, we believe that currently there is no need since we have discussed architecture components of existing works in table 6 and section 5.3, and we will be covering this in our future work. Thanks for your comments.

Reviewer 3 Report

This paper survey the use of the state-of-the-art computing technologies in agriculture. This is a well written, easy to follow and interesting paper. However, I have following comments that need to be addressed before the paper is accepted.

1) In section 2, the authors must refine the presentation of the current state of the related survey papers, their research fields and the issues/limitations should be put into the context. The purpose of the literature survey is to highlight for the involved referenced papers the main contribution that the authors of the referenced papers have brought to the current state of knowledge, the methods used by the authors of the referenced papers, a brief presentation of the main obtained results and some limitations of the referenced article to create the research gap. 

2) What is the academic contribution of this survey? Please also refine the presentation of key contributions of the paper in Introduction. 

3) Please specify key insights and research limitations in the conclusions section.

Author Response

Points:

This paper survey the use of the state-of-the-art computing technologies in agriculture. This is a well written, easy to follow and interesting paper. However, I have following comments that need to be addressed before the paper is accepted.

1) In section 2, the authors must refine the presentation of the current state of the related survey papers, their research fields and the issues/limitations should be put into the context. The purpose of the literature survey is to highlight for the involved referenced papers the main contribution that the authors of the referenced papers have brought to the current state of knowledge, the methods used by the authors of the referenced papers, a brief presentation of the main obtained results and some limitations of the referenced article to create the research gap. 

Response: We hope you were referring to Section 3. We have refined and added some points based on your comments in the last two paragraphs.

2) What is the academic contribution of this survey? Please also refine the presentation of key contributions of the paper in the Introduction. 

Response: We have refined the last paragraph of the introduction section based on your comments.

3) Please specify key insights and research limitations in the conclusions section.

Response: We have refined both paragraphs of the conclusion section with research limitations and key insights.

Reviewer 4 Report

This survey paper aims to provide a systematic literature review of current works that have been done in Cloud, Fog, and Edge Computing applications in the smart agriculture domain from 2015 until now. Authors searched and reviewed 55 relevant research on new computing paradigms with smart agriculture. Furthermore, they analyzed and discussed the agricultural application domains, research approaches, the application of used combinations, the challenges of smart agriculture, and future research directions. Authors also proposed a new architecture model with the combinations of Cloud-Fog-Edge.

Author used the Preferred Reporting Items for Systematic Reviews and Meta-Analyses to conduct this systematic review. They try to investigate and provide a review of existing research on Cloud, Fog, and Edge Computing applications in the agricultural field. The conclusion is based on systematic analyses and presented properly. The language of this paper is well written and meets the requirements of a scientific paper.

Author Response

Comments:

This survey paper aims to provide a systematic literature review of current works that have been done in Cloud, Fog, and Edge Computing applications in the smart agriculture domain from 2015 until now. Authors searched and reviewed 55 relevant research on new computing paradigms with smart agriculture. Furthermore, they analyzed and discussed the agricultural application domains, research approaches, the application of used combinations, the challenges of smart agriculture, and future research directions. Authors also proposed a new architecture model with the combinations of Cloud-Fog-Edge.

Author used the Preferred Reporting Items for Systematic Reviews and Meta-Analyses to conduct this systematic review. They try to investigate and provide a review of existing research on Cloud, Fog, and Edge Computing applications in the agricultural field. The conclusion is based on systematic analyses and presented properly. The language of this paper is well written and meets the requirements of a scientific paper.

Response: Thank you for your comments.

Round 2

Reviewer 3 Report

The paper is improved after revision. Hence, my decision is to Accept in the present form.